# Active Fault-Locating Scheme for Hybrid Distribution Line Based on Mutation of Aerial-Mode Injected Pulse

Zhuang Jiang, Xiangjun Zeng, Feng Liu *, Kun Yu, Lanxi Bi and Youpeng Wang

State Key Laboratory of Disaster Prevention & Reduction for Power Grid, Changsha University of Science and Technology, Changsha 410114, China; eezjiang@163.com (Z.J.)
* Correspondence: eefengliu@163.com

**Abstract:** Due to the overlap of initial traveling wave signals, the traveling wave propagation process in hybrid distribution lines is complicated to analyze. The most significant challenge posed by the traditional passive traveling wave-locating method for hybrid distribution lines lies in identifying the fault section and distinguishing the reflected wave from the fault point or the hybrid connection points. Based on this approach, with the application of the aerial-mode component of the pulse signal generated at the fault point, a fault-section-identification and fault-locating scheme for hybrid distribution feeders with active pulse injection is proposed. When power in a line is cut after a single-to-line ground (SLG) fault occurs, the same pulse is injected into the three phases from the neutral point of the coupling capacitor bank to construct the zero-mode component, which propagates to the SLG fault three-phase asymmetrical point, producing an aerial-mode component that is reflected back to the first end of the line. With the application of the arrival time of an aerial-mode wavefront, it is simple to locate the SLG fault for arbitrary forms of hybrid lines. The simulation results confirm the feasibility of the fault-locating scheme under different feeders, different fault locations, and fault resistances. The results of the experiments confirm the high practical value of the proposed method.

**Keywords:** hybrid distribution line; pulse injection; aerial-mode traveling wave; zero-mode traveling wave; active fault location

## 1. Introduction

Affected by urban planning and other factors, overhead-cable hybrid distribution lines are increasingly used in distribution networks, and complex-terrain hybrid distribution lines are susceptible to adverse weather and construction activities, leading to permanent grounding faults and high maintenance costs [1,2]. Therefore, it is necessary to study precise and effective fault-location methods to help maintenance personnel find fault points and restore power supplies [3,4].

The fault-location methods for hybrid lines fall into two main categories: the impedance-based method and the travelling-wave-based method [5–9]. Travelling-wave-based methods offer higher reliability and precision compared to impedance-based ones. For two-section hybrid lines, a travelling-wave-based method that considers the arrival times of various reflected fault waves on each line segment is proposed in [10]. However, detecting reflected wave signals in multisection lines is challenging. To address this problem, the study presented in [11] involves comparing the arrival times of initial fault waves with the calculated wave arrival time before the fault occurs. A faulted-section-identification method for hybrid line is proposed in [12], using the ratios of initial magnitudes are used to identify the faulted section effectively and accurately.

In [13], fault location for hybrid lines is achieved by utilizing the time–frequency characteristics of traveling waves through the complete waveform. With the advancement of intelligent algorithms, travelling-wave-based methods can be integrated with advanced techniques to identify faulty sections in hybrid lines. The identification of faulty line sections

in hybrid lines was realized by using the transient energy characteristics of fault signals, based on a support vector machine (SVM) classifier and a time–time (TT) transformation method [14]. Additionally, fault characteristics of zero-mode and aerial-mode signals are analyzed with principal components analysis and Euclidean distance [15]. However, their reliability hinges on diverse training data, requiring updates for network configuration changes [16,17].

The accuracy of the above-mentioned passive traveling wave fault-location methods may be adversely affected by the original acquisition of the fault traveling wave characteristic signals. Elkalashy et al. [18] used thyristors to generate identical disturbance signals from the neutral point in in all three phases of the distribution lines. Qi et al. [19] implemented fault distance measurement by connecting resistors in parallel to the neutral point to generate interference signals. However, these methods often directly inject signals from medium-voltage lines, typically at the beginning of the distribution network, leading to operational complexity and safety risks. Signal amplitude reduction caused by branch lines may result in difficulties in identifying reflected signals and inaccurate fault distance measurements at the rear end of the distribution network. A method was proposed in [20] for injecting pulses from the low-voltage side of a distribution transformer to locate faults. This method addresses the issue of detecting small signal amplitudes at the end of the line. However, the chosen signal injection position is near the user side, resulting in increased implementation complexity.

For accurate fault location, the detection of the arrival time of traveling wave signals is crucial. The wavelet transform (WT) method is commonly employed to handle transient signals and the first significant peak of the traveling wave signals. However, the detection outcomes of wavelet transform can be significantly influenced by both the choice of the wavelet basis function and the decomposition scale. To address the problem of wavelet transform, Wang et al. [21] utilize the Hilbert–Huang transform (HHT) approach to detect the wavefronts and accomplish fault location. HHT, based on empirical mode decomposition (EMD) and Hilbert transform, offers advantages over traditional methods as it is not constrained by wavelet functions or decomposition scales. Furthermore, it can effectively represent the instantaneous frequency of time-varying signals. However, one notable drawback is the potential for mode aliasing in the EMD process when decomposing signals. To address the modal mixing in EMD, a multibranch, small-current grounding system-location method based on VMD and Teager energy operator (TEO) calibration was proposed in [22,23]. However, the decomposition effect of VMD is affected by the amount of decomposition.

In this paper, a novel method for accurately identifying fault sections and achieving fault location in multisection hybrid lines is proposed. The influence of attenuation and dispersion factors on the distribution of pulse energy was investigated, and a high-frequency energy factor was defined based on this, providing criteria for selecting pulse signals. The high-voltage pulse is injected into the three-phase neutral point of the coupling capacitor from the first end of the hybrid line. According to the propagation characteristics of the voltage pulse at the distribution hybrid line and the single-phase grounding fault point, the time difference between the zero-mode and the aerial-mode signal arrivals at the first-end detection device was utilized to identify fault segments. After identifying fault segments, a single-end fault-location algorithm for arbitrary hybrid lines was developed. To accurately detect the arrival time of initial fault wave signals, the VMD–information entropy method and the Teager energy operator were employed in a novel wavefront-detection approach.

## 2. Propagation Characteristics of Pulse Signals in Hybrid Distribution Lines

### 2.1. Dispersion Characteristics of Pulse Signals

According to the principles of distribution line theory, when the frequency of the transmitted signal in the cable is high, it is necessary to employ a distributed parameter model for simulating the electrical structure of the cable. In Figure 1, the symbols $R_0$, $L_0$, $G_0$, and $C_0$ represent the resistance, inductance, conductance, and capacitance per unit length of the cable,

respectively. The specific values of these parameters are dependent on various factors, such as the cable's structure, materials, dimensions, and other relevant characteristics.

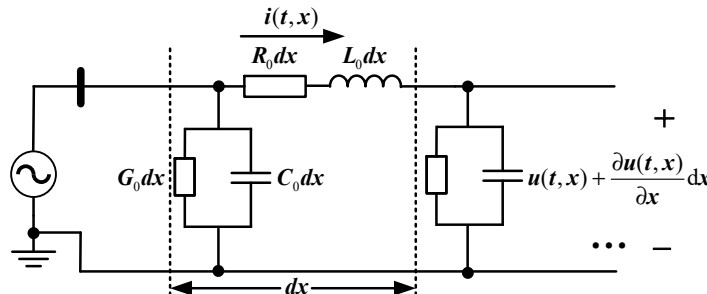

**Figure 1.** Micro-element equivalent circuit for single-phase line.

The partial differential equation system can be formulated to yield the following:

$$-\frac{\partial u(t,x)}{\partial x} = R_0 i(t,x) + L_0 \frac{\partial i(t,x)}{\partial x},$$

$$-\frac{\partial i(t,x)}{\partial x} = G_0 u(t,x) + C_0 \frac{\partial u(t,x)}{\partial x}. \tag{1}$$

Equation (1) enables the Laplace transformation of the pulse signal, comprising multiple frequency components, expressed as:

$$-\frac{dU(s,x)}{dx} = (R_0 + L_0 s) \cdot I(s,x) = Z_0 \cdot I(s,x),$$

$$-\frac{dI(s,x)}{dx} = (G_0 + C_0 s) \cdot U(s,x) = Y_0 \cdot U(s,x). \tag{2}$$

where $Z_0$ is the unit-length series impedance and $Y_0$ is the unit-length shunt admittance. Rearranging the equation yields:

$$\frac{dU^2(s,x)}{dx^2} - Z_0 Y_0 \cdot U(s,x) = 0. \tag{3}$$

The general solution for the voltage pulse signal can be obtained as follows:

$$U(s,x) = A \cdot e^{-\sqrt{Z_0 Y_0} x} + B \cdot e^{\sqrt{Z_0 Y_0} x}. \tag{4}$$

where the constants $A$ and $B$ are undetermined constants determined by the boundary conditions at the load end of the line. The propagation coefficient can be defined as follows: $\lambda(s) = \sqrt{Z_0 Y_0}$. By substituting $s = j\omega_i$, we can obtain the following expression [24]:

$$\lambda(j\omega_i) = \alpha(\omega_i) + j\beta(\omega_i). \tag{5}$$

In Equation (5), the real part can be used to determine the rate at which the amplitude of the frequency components in the pulse signal attenuates. Moreover, the imaginary part can be used to determine the rate at which the phase of the frequency components in the pulse signal shifts, serving as a phase offset function [25]. The relationship between the wave velocity and phase shift coefficient $\beta(\omega_i)$ for a single frequency can be determined by analyzing the equation of motion of the equiphase plane of each frequency component of the voltage pulse signal [26]:

$$v_m(\omega_i) = \frac{\omega_i}{\beta(\omega_i)}. \tag{6}$$

Equation (6) shows that the propagation velocities of different frequency components within the pulse signal vary, with higher frequencies propagating at a faster rate. Consequently, the pulse signal experiences frequency offset and dispersion distortion after a certain period of propagation. This phenomenon results in the rising edge of the pulse, predominantly consisting of higher-frequency components, while the trailing edge in-

cludes both medium- and low-frequency components. The pulse propagating in the line of each mode component is composed of different frequency components. According to Equation (6), as the frequency of the propagation velocity increases, the pulse propagates faster through the medium, leading to distortion and reducing accuracy in determining the pulse's arrival time.

### 2.2. Analysis of Transient Processes of Injected Pulses at the SLG Fault Point

In the transient analysis process of the three-phase system in the distribution network, uncoupled moduli are typically employed to analyze the wave processes on the line. Establishing and solving the equations for line node potentials are more conveniently computed in the phasor domain, achieved through conversion using the Karrenbauer phase-mode transformation matrix [27]:

$$
\begin{bmatrix} x_0 \\ x_1 \\ x_2 \end{bmatrix} = \frac{1}{3} \begin{bmatrix} 1 & 1 & 1 \\ 1 & -1 & 0 \\ 1 & 0 & -1 \end{bmatrix} \begin{bmatrix} x_a \\ x_b \\ x_c \end{bmatrix}.
\tag{7}
$$

where $x_a$, $x_b$, and $x_c$ represent the phases and $x_0$, $x_1$, and $x_2$ represent the moduli.

When the same pulse signal is injected in all three phases, the following expression can be obtained: $u_a = u_b = u_c = u_i$. Here, $u_i$ is the initial amplitude of the injected pulse. According to the above transformation equation, the modulus voltage signal can be obtained as $u_0 = u_i$, $u_1 = 0$, $u_2 = 0$. We let the zero-mode and aerial-mode wave impedances of the line be $Z_0$ and $Z_1$, respectively, and the fault transition resistance be $R_f$. At the SLG fault point, by using the magnitude relationship among the incident pulse signal, the inverse traveling pulse signal, and the forward-traveling pulse signal, and according to the boundary conditions of the fault point, the following expressions can be obtained:

$$
\begin{cases} u_{0f1} = -\dfrac{Z_0 u_{0q1}}{Z_0 + 2Z_1 + 6R} - \dfrac{Z_0(u_{1q1} + u_{2q1})}{Z_0 + 2Z_1 + 6R} \\ u_{1f1} = -\dfrac{Z_1 u_{1q1}}{Z_0 + 2Z_1 + 6R} - \dfrac{Z_1(u_{0q1} + u_{2q1})}{Z_0 + 2Z_1 + 6R} \\ u_{2f1} = -\dfrac{Z_1 u_{2q1}}{Z_0 + 2Z_1 + 6R} - \dfrac{Z_1(u_{0q1} + u_{1q1})}{Z_0 + 2Z_1 + 6R} \end{cases}.
\tag{8}
$$

where $u_{0f1}$, $u_{1f1}$, and $u_{2f1}$ are the inverse pulse voltage waves of the zero mode, the aerial mode on the upstream side of the fault point, and the reflected traveling wave of the voltage of each mode at the fault point; $u_{0q1}$, $u_{1q1}$, and $u_{2q1}$ are the forward-traveling voltage waves of the zero mode, the aerial mode on the upstream side of the fault point, and the incident traveling wave of the voltage of each mode at the fault point.

When the pulse propagates to the fault point, the following expressions can be obtained:

$$
\begin{cases} u_{0q1} = u_0 = \delta_0 \beta_0 u_i, \\ u_{1q1} = u_1 = 0, \\ u_{2q1} = u_2 = 0. \end{cases}
\tag{9}
$$

where $\delta_{zm}$ is the zero-mode attenuation coefficient and $\beta_{zm}$ is the zero-mode refraction coefficient. Substituting Equation (9) into Equation (8) yields the following:

$$
\begin{cases} u_{0f1} = -\dfrac{Z_0 \delta_{zm} \beta_{zm} u_i}{Z_0 + 2Z_1 + 6R} \\ u_{1f1} = -\dfrac{Z_1 \delta_{zm} \beta_{zm} u_i}{Z_0 + 2Z_1 + 6R} \\ u_{2f1} = -\dfrac{Z_1 \delta_{zm} \beta_{zm} u_i}{Z_0 + 2Z_1 + 6R} \end{cases}.
\tag{10}
$$

After the reflection at the fault point, an aerial-mode pulse wave with the opposite polarity to the incident zero-mode voltage wave appears in the circuit. Moreover, it can be observed that the amplitude of the reflected aerial-mode voltage pulse wave is directly

proportional to the amplitude of the incident zero-mode voltage pulse and inversely proportional to the magnitude of the ground resistance.

Given that the zero-mode is influenced by the zero-sequence inductance and resistance, it is important to note that these parameters are intricately tied to frequency, owing to the skin effect observed in the line–earth return. The zero-sequence inductance experiences a notable decrease with increasing frequency, while the zero-sequence resistance undergoes a significant increase with the rise in frequency. It can be seen from Equations (5) and (6) that $\alpha$ and $V_m$ of the zero-mode increase sharply with an increasing frequency, and the attenuation and phase shift of the zero-mode component in the propagation of the pulsed signal are the most significant. Moreover, the higher the frequency of the zero-mode component, the more serious the attenuation will be in the propagation process. The aerial mode is affected by the inductance and resistance of the positive sequence, but the parameters of the positive sequence's resistance and inductance are much less affected by the change in frequency. Thus, $\alpha$ and $V_m$ of the aerial-mode increase with frequency, but they are much less affected by the frequency than the zero mode is. Therefore, although $\alpha$ and $V_m$ of the aerial mode also increase with frequency, they are much less affected by frequency than those of the zero mode are. As the attenuation of the aerial-mode traveling wave is small, the aerial-mode wave propagation velocity with frequency also has less influence. Therefore, in this paper, we mainly take the aerial-mode traveling wave as an example to consider the problem of determining the wavefront arrival time.

Based on the above analysis, the location algorithm in this paper takes the moment of the detected aerial-mode voltage pulse wavefront as the moment of arrival via fault reflection, which is the moment of arrival of the highest-frequency pulse component.

## 3. Propagation Characteristics of Pulse Signals in Hybrid Distribution Lines

### 3.1. Three-Phase Coupling Injection of Voltage Pulses

In response to the challenges posed by fixed signal injection positions and the difficult identification of reflected waves at fault points in fault location for hybrid distribution lines within distribution networks, we propose a novel method. This method entails fault location for hybrid distribution lines through pulse injection, achieved by analyzing the propagation characteristics of voltage pulse signals within such lines. The voltage pulses are generated by the fault-locator device and then injected into the distribution system via an integrated capacitive coupler. During operation at power frequency, the capacitor functions as an insulation element between the network and the fault-locator device. However, when operating at high frequencies, it transforms into a short circuit, facilitating direct connection between the fault-locator device and the distribution system.

The integrated capacitive coupler efficiently isolates the line from the pulse-injection device during power frequency electrical conditions, ensuring safety for personnel and equipment while having no impact on the line's regular operation. The pulse signal is injected from either the neutral point of the line's reactive compensation capacitor or a self-constructed capacitor bank. In instances where the pulse width of the injected pulse signal falls within the microsecond range, the signal bandwidth can exceed 100 kHz, and the capacitor can efficiently couple the pulse into the line with minimal loss. In comparison to the signal-injection method employed in traditional offline fault location approaches, the signal-injection method proposed in this paper offers enhanced flexibility and reliability, allowing for pulse signal injection from any substation along the line, thus ensuring ease of implementation.

### 3.2. Selection of Injected Pulse Waveforms

Building upon the aforementioned analysis of pulse-propagation characteristics, we establish a foundation for selecting the appropriate form of pulse injection. The dispersion of pulse signals in transmission lines is particularly pronounced due to the distinctive characteristics of overhead lines and cable structures. To facilitate the subsequent feature extraction of the pulse signal, it is crucial to ensure a broad pulse spectrum. This ensures that the high-frequency components within the pulse wavefront attain their maximum phase

velocity. Therefore, the dispersion cutoff frequency, $f_{th}$, is defined as the signal frequency at which the phase velocity reaches the maximum phase velocity. The specific calculation process of the dispersion cutoff frequency, $f_{th}$, can be found in Appendix A [28,29].

However, in engineering applications, due to the equipment sampling accuracy, measurement range, signal-to-noise ratio, and other factors, when traveling wave data are processed, low-frequency signals cannot be completely filtered out, and signals above the $f_{th}$ band are difficult to measure. Therefore, the injected pulse signal should be injected in such a way that the pulse signal is measurable, and so that the injected signal does not overlap with the reflected signal. Moreover, the injected pulse amplitude should be as large as possible, and the frequency should be as high as possible.

To establish a basis for selecting injection pulse parameters, this paper analyzes the spectrum of the injected pulse signal. Frequency components in pulse signals higher than $f_{th}$ can be approximately considered to exhibit no dispersion. Let the pulse signal be represented as $A_0 f(t)$, where $A_0$ is the initial amplitude of the injected pulse and $\sigma$ is the width of the pulse signal. The fast Fourier coefficient expression of the pulse signal is as follows (the specific calculation process can be found in Appendix B):

$$X_k(w, A_0, \sigma) = \frac{1}{\sigma} \int_0^{\sigma} A_0 f(t) e^{-jwt} dt. \tag{11}$$

Generally, as the pulse width $\sigma$ increases, the change in the exponential term becomes slower, indicating that lower-frequency components will be included in the spectrum. Conversely, when the pulse width $\sigma$ decreases, the change in the exponential term becomes faster, and higher-frequency components will be included in the spectrum. Equation (11) shows that the spectrum $|X_k(w, A_0, \sigma)|$ of the pulse signal is related to the amplitude and width of the pulse signal; the narrower the pulse width is, the wider the pulse spectrum. The absolute value of the pulse spectrum is taken and integrated over the frequency, and the integral value is treated as the spectrum energy value of the pulse in a certain frequency band. The high-frequency component coefficient $\kappa$ is defined to characterize the proportion of high-frequency components in the pulse signal, and the value of the coefficient is determined by the pulse shape, amplitude, and width, where the molecular part represents the high-frequency spectrum energy and the denominator part represents the full-frequency domain spectrum energy. The criterion can be set to select the injected pulse width:

$$\kappa(A_{\max}, \sigma_{\min}) = \frac{\int_{f_{th}}^{\frac{f_s}{2}} |X_k(w, A, \sigma)| df}{\int_0^{\frac{f_s}{2}} |X_k(w, A, \sigma)| df} \le \lambda. \tag{12}$$

where $f_s$ is the sampling frequency and $A_{\max}$ refers to the maximum magnitude of the pulse signal that can be injected. It is important to avoid setting the pulse amplitude too high, which can result in damage to the insulation of the line or can exacerbate the fault condition. Thus, it is usually taken as the voltage level of the distribution line being injected. $\sigma_{\min}$ represents the minimum width of the injected pulse; $\lambda$ is the criterion setting value, which is dependent on the actual working conditions and normally set to 0.5.

Due to the variation in the propagation velocity among the frequency components within the pulse signal, the high-frequency components in the pulse signal reach the measurement point before the low-frequency components do. The propagation velocity of the fault traveling wave depends on the wave velocity associated with the high-frequency components during the pulse rise time. Selecting a suitable pulse shape is crucial for determining accurate fault locations, and it is preferable to use pulse signals that encompass a broader frequency range in the frequency domain. Considering the pulse signal parameters, $A_0 = 5\,\text{kV}, \sigma = 2\,\mu\text{s}$, the frequency spectra of three types of pulse signals with identical pulse widths and amplitudes are obtained. Figure 2 demonstrates that the rectangular pulse has a broader frequency distribution in the frequency domain and is comparatively less susceptible to the wave velocity and signal attenuations than the other two pulse

shapes. Considering the cost of signal generation, the square, wave-like signal generated by switching has a low hardware cost, which is favorable for the promotion and application of this technology. Consequently, the rectangular pulses are used as the injected pulse signals in this paper.

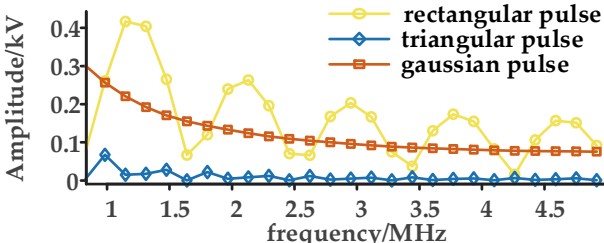

**Figure 2.** Common pulse signal spectrum.

### 3.3. Fault-Location Algorithm Based on the Aerial-Mode Pulse Arrival Time Difference Sequence

To address the challenge of accurately locating fault points in a distribution network with branch feeder lines using a single device, we propose the utilization of an auxiliary detection device positioned at the end of each branch feeder. This auxiliary device assists in selecting the faulty branch feeder. When the zero-mode pulse signal reaches the fault point, an aerial-mode signal is generated, propagating to both ends of the line. The auxiliary detection device at the end of the branch feeder then detects the aerial-mode pulse signal. In contrast, non-faulty lines only exhibit zero-mode signals.

After the fault feeder is selected, the fault location can be determined by using the time difference between the zero mode and the aerial mode. In the actual distribution grid, there are two main types of connections for hybrid overhead-cable lines: type-A and type-B. A type-A hybrid line consists of an overhead line and a cable. A type-B line consists of two overhead lines and a cable, i.e., the overhead line–cable–overhead line form, which are more commonly observed in distribution networks.

It is worthwhile to illustrate this fault-location method with the example of a type-B hybrid line, as shown in Figure 3; here, $MX_1$ and $X_2N$ are overhead lines, $X_1X_2$ is a cable line, and $X_1$ and $X_2$ represent the connection points for the overhead line and cable, respectively. Due to the different parameters of the two lines, the propagation velocity is also different. We let the zero-mode and aerial-mode wave velocities of the overhead line be $v_0$ and $v_1$, respectively, and the zero-mode and aerial-mode wave velocities of the cable line be $vc_0$ and $vc_1$, respectively. The length of the cable line is denoted by $L_{X1X2}$, while $L_{MX1}$ and $L_{X2N}$ indicate the lengths of the two sections of the overhead line.

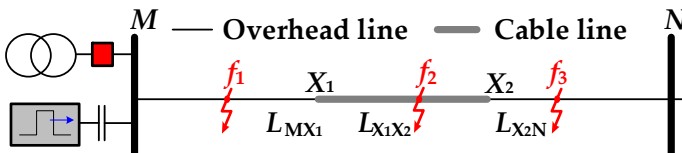

**Figure 3.** Model of the type-B hybrid distribution line.

If the structure and parameters of the line are known, then it becomes possible to calculate the time required for both zero-mode and aerial-mode signals to propagate through each section of the line. Consequently, if the time difference ($\Delta t$) between the pulse signal's arrival at the fault point and its return to the first-end detection device is known, along with the specific line structure, then the precise location of the fault point can be determined. Based on this information, a fault-location algorithm is proposed for the hybrid line.

First, the determination of the fault section is achieved by analyzing the time difference between the arrival of the zero-mode and aerial-mode signals at the first-end detection de-

vice. Then, the fault location is determined using the specific time difference corresponding to the fault section.

When the fault points are located at $X_1$, $X_2$, and $N$, the times needed for the aerial-mode signal to reach the first-end detection device are $\Delta t_1$, $\Delta t_2$, and $\Delta t_3$, respectively:

$$\begin{cases} \Delta t_1 = \frac{L_{MX_1}}{v_0} + \frac{L_{MX_1}}{v_1} \\ \Delta t_2 = \frac{L_{MX_1}}{v_0} + \frac{L_{MX_1}}{v_1} + \frac{L_{X_1X_2}}{v_{c0}} + \frac{L_{X_1X_2}}{v_{c1}} \\ \Delta t_3 = \frac{L_{MX_1}}{v_0} + \frac{L_{MX_1}}{v_1} + \frac{L_{X_1X_2}}{v_{c0}} + \frac{L_{X_1X_2}}{v_{c1}} + \frac{L_{X_2N}}{v_0} + \frac{L_{X_2N}}{v_1} \end{cases} \tag{13}$$

$$\begin{cases} L_{f1} = \frac{v_0v_1}{v_0+v_1}\Delta t, 0 < \Delta t < \Delta t_1, \\ L_{f2} = \frac{v_{c0}v_{c1}}{v_{c0}+v_{c1}}(\Delta t - \Delta t_1) + L_{MX_1}, \Delta t_1 < \Delta t < \Delta t_2, \\ L_{f3} = \frac{v_0v_1}{v_0+v_1}(\Delta t - \Delta t_2) + L_{MX_1} + L_{X_1X_2}, \Delta t_2 < \Delta t < \Delta t_3. \end{cases} \tag{14}$$

Fault section identification is achieved by comparing magnitudes of $\Delta t$, $\Delta t_1$, $\Delta t_2$, and $\Delta t_3$:

1. When $0 < \Delta t < \Delta t_1$, the fault is located in the $MX_1$ overhead line section.
2. When $\Delta t_1 < \Delta t < \Delta t_2$, the fault occurs in the $X_1X_2$ cable section.
3. When $\Delta t_2 < \Delta t < \Delta t_3$, the fault occurs in the $X_2N$ overhead line section.

The fault location formula for a type-B hybrid line is as follows:

The analysis described above, in conjunction with Equations (9) and (10), can be applied to any type of hybrid lines. If the fault is situated in the $X_{k-1}X_k$ overhead section, then the distance between the fault point and the injection point can be determined as follows:

$$L_f = \frac{v_0v_1}{v_0 + v_1}(\Delta t - \Delta t_k) + \sum_{i=1}^{k-1} L_i, \Delta t_{k-1} < \Delta t < \Delta t_k. \tag{15}$$

If the fault is located in the $X_{k-1}X_k$ cable section, then the distance between the fault point and the injection point can be calculated as follows:

$$L_f = \frac{v_{c0}v_{c1}}{v_{c0} + v_{c1}}(\Delta t - \Delta t_k) + \sum_{i=1}^{k-1} L_i, \Delta t_{k-1} < \Delta t < \Delta t_k. \tag{16}$$

*3.4. A Wavefront Calibration Method Based on VMD-IE-NTEO*

To mitigate the impact of interference noise on fault location outcomes and precisely calibrate the arrival time of aerial-mode signal wavefronts, a novel method is proposed. Initially, the variational mode decomposition (VMD) method employs a variational problem to achieve an optimal result and dynamically decomposes multicomponent signals by iteratively adjusting the central frequency and bandwidth of each mode function. This approach yields a set of intrinsic mode functions (IMFs); these are characterized by finite bandwidths that closely envelop their respective central frequencies.

Given that the pulsed traveling wave signal encompasses a wide-band signal, and the noise signal overlaps with the high-frequency IMF components of the traveling wave, significantly hindering the identification of the traveling wavefront, it becomes essential to filter out the noise-polluted signal components to the fullest extent possible. To achieve this, information entropy (IE) is introduced as an index that can effectively gauge the level of noise present in a signal. The greater the signal uncertainty is, the greater the IE value [30]. The IE is used to calculate the noise level of each IMF, and the optimal IMF component is selected to represent the mutational properties of the original signal. If the IE of an IMF significantly exceeds the IE of the original signal, then the IMF component is considered to be heavily contaminated by noise and needs to be eliminated. In this paper, a new parameter is introduced to the conventional TEO algorithm. This parameter, which is denoted as $j$, can be used to calculate the energy values by considering $j$ points both before and after, effectively mitigating the impact of noise on the extraction of the pulse signal wavefront.

$$\begin{cases} \phi[s(n)] = s^2(n) - s(n+j)s(n-j) \\ |\psi(n)| = \dfrac{2\phi[s(n)]}{\sqrt{\phi[s(n+1)] - \phi[s(n-1)]}} \end{cases} \tag{17}$$

where $\phi[s(n)]$ represents the energy operator, $\psi(n)$ represents the energy operator envelope curve, and $s(n)$ represents a discrete signal sequence.

The steps for implementing fault location when a single-line-to-ground (SLG) fault occurs in a hybrid distribution line are depicted in Figure 4, based on the aforementioned principle of fault location.

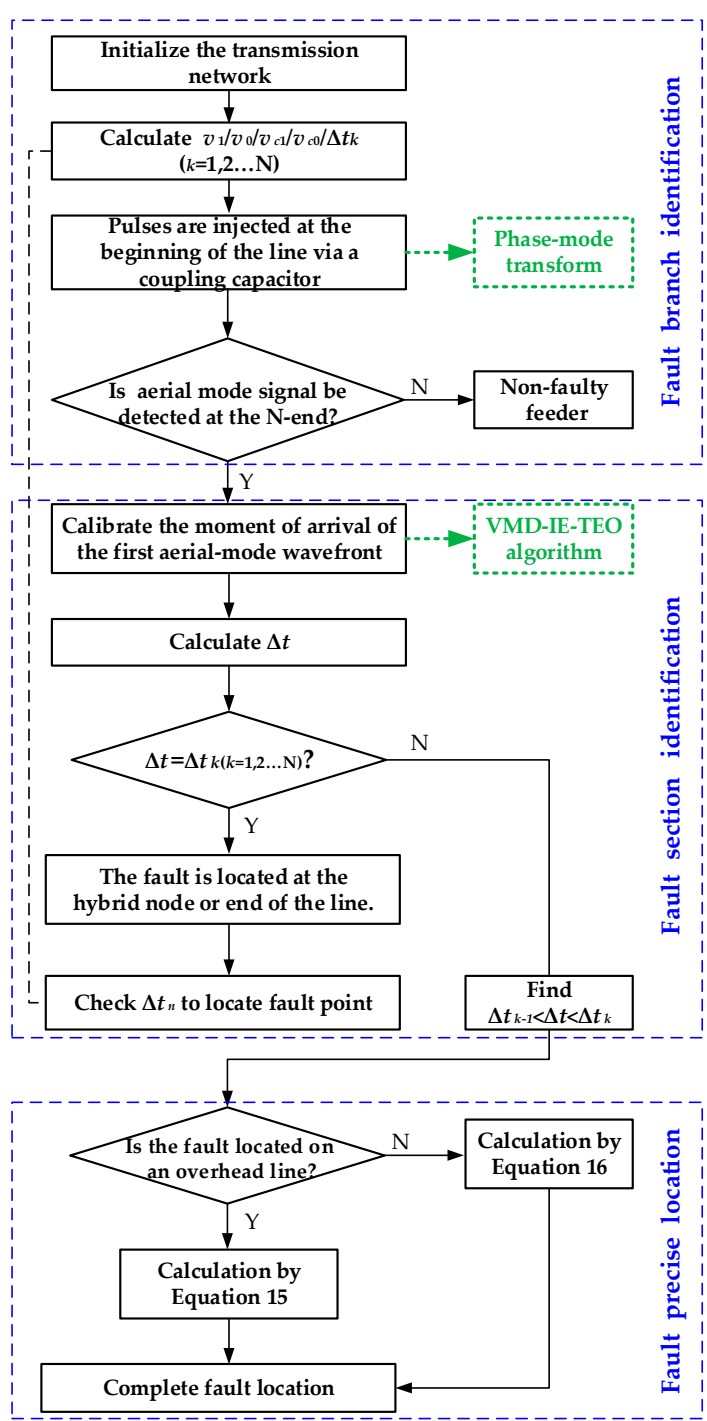

**Figure 4.** Proposed fault-locator algorithm for hybrid distribution line.

## 4. Simulation Analysis

A 10 kV hybrid distribution network model is constructed as a simulation project by means of PSCAD, as depicted in Figure 5. The network included a branch consisting of a type-A 8.5 km hybrid feeder line and a type-B 15 km hybrid feeder line, and the line parameters are shown in Table 1. After further calculations based on the relevant hybrid line parameters, the maximum cutoff frequency, $f_{th}$, for the hybrid line is 45 kHz. To ensure the insulation integrity of the hybrid line remains uncompromised, the amplitude of injected pulses should be selected at one-tenth of the voltage rating value $A_{max} = 1$ kV. According to the pulse width selection criterion, Equation (12), the minimum width of the injected pulse is $\sigma_{min} = 1.95$ µs (approximately rounded to 2 µs). A voltage pulse-injection device was positioned at the M-end of the hybrid line. Synchronized pulse-signal-detection devices were installed at both the M-end and the N-end of the hybrid line to capture pulse signals at a sampling frequency of 10 MHz.

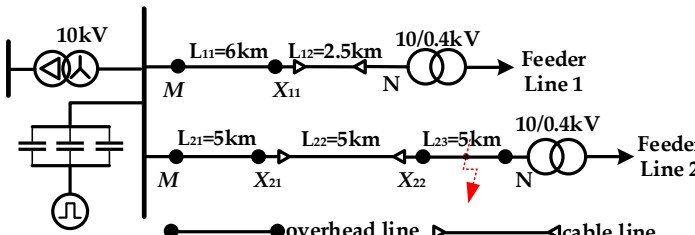

**Figure 5.** Diagram of the simulation model.

**Table 1.** Distribution network model line parameters.

| Line Type | Phase Sequence | *R* (Ω/km) | *C* (µF/km) | *L* (mH/km) |
|---|---|---|---|---|
| Overhead line | Positive | 0.1700 | 0.0097 | 1.1934 |
| | Zero | 0.2300 | 0.0080 | 2.4309 |
| Cable Line | Positive | 0.2700 | 0.3390 | 2.4717 |
| | Zero | 2.7000 | 0.2800 | 3.8226 |

Based on the hybrid line parameters in Table 1, the zero-mode and aerial-mode pulse-propagation velocities corresponding to overhead lines and cable lines can be calculated by $1/\sqrt{LC}$ as follows:

$$\begin{cases} v_0 = 253.5\text{m}/\text{µs} \\ v_1 = 297.9\text{m}/\text{µs} \end{cases}, \begin{cases} v_{c0} = 158.7\text{m}/\text{µs} \\ v_{c1} = 187.7\text{m}/\text{µs} \end{cases} \tag{18}$$

### 4.1. Tests for Faulty Feeder Selection

An SLG fault (single line to ground fault) is assumed to occur in feeder line $L_2$, with a fault resistance ($R_f$) of 100 Ω and situated 12.5 km away from the first end of the line. For sampled aerial-mode signals at the N-end, it is observed that the line mode signal is detected at the N-end in feeder $L_2$, while no signal is detected at the N-end in feeder $L_1$. This is because the zero-mode pulse signal reaches the fault point, consequently generating an aerial-mode signal. Subsequently, the aerial-mode pulse signal propagates to both ends of the faulty feeder. Due to the transmission attenuation of the aerial-mode pulse signal, as depicted in Figure 6, the non-faulty feeder is almost unable to detect the aerial-mode pulse signal, facilitating the selection of the faulty feeder.

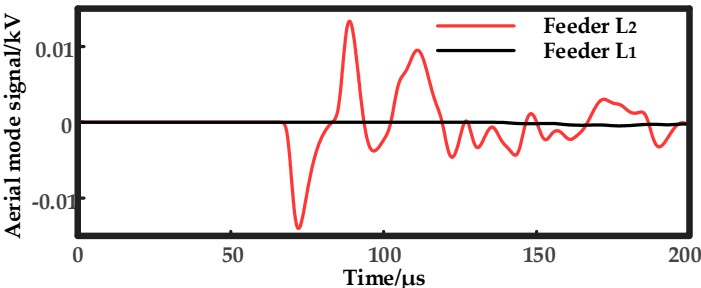

**Figure 6.** Aerial-mode pulse waveform at the M-end detection point.

*4.2. Tests for VMD-IE-TEO Method*

To accurately calibrate the wavefront arrival moment, the simulation data for MD-IE-TEO were selected before 0.4 ms. Furthermore, we intended to study the influence of white noises for pulse-wavefront-detection methods. When white noise signal is added in the sampled aerial-mode signal, the wavefront signal is submerged by the noise and difficult to detect. The aerial-mode signal with 15 dB Gaussian noise and its corresponding spectrum are shown in Figure 7.

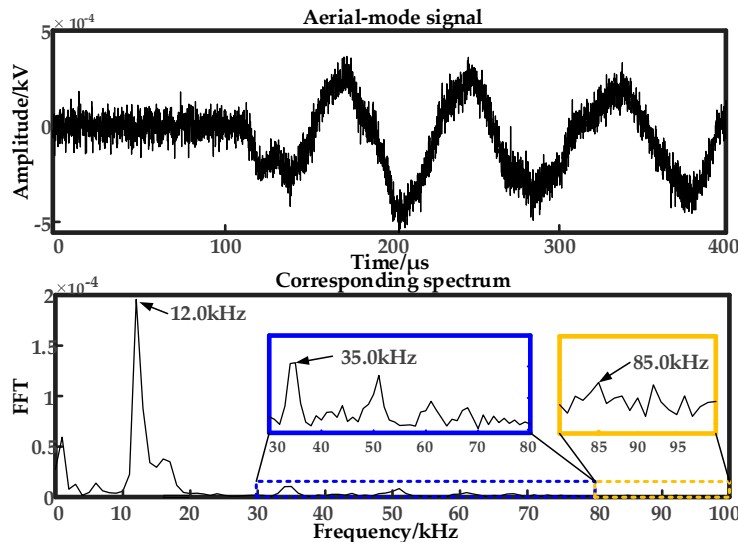

**Figure 7.** Aerial-mode pulse waveform and corresponding spectrum.

The frequency domain of the original aerial-mode pulse signal is divided into three parts: low–medium (0–30 kHz), medium (30–80 kHz), and high-frequency (80–100 kHz) bands. The frequencies of the corresponding dominant-frequency components in each frequency band are 12.0 kHz, 35 kHz, and 85 kHz.

After the fault occurs, the sampled aerial-mode pulse signal undergoes decomposition using the VMD algorithm, with the number of IMF components set to $k = 5$ and the penalty factor being $\alpha = 2000$. The resulting IMFs and their corresponding spectra are displayed in Figure 8. The VMD algorithm decomposes the various modes into the following components: IMF1, representing the low-frequency component; IMF2, representing the second-lowest-frequency component; IMF3, representing the medium-frequency component; IMF4, representing the second-highest-frequency component; IMF5, representing the highest-frequency component. IMF1 and IMF2 capture the overall trend of the original aerial signal, while IMF3, IMF4, and IMF5 capture the mutation information of the original aerial signal.

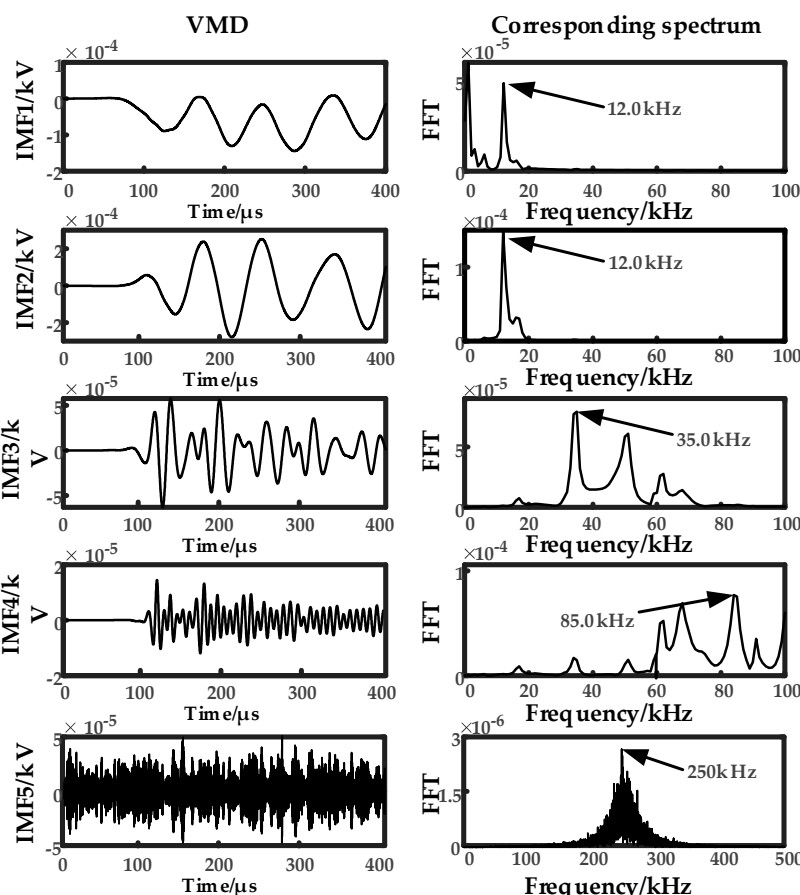

**Figure 8.** VMD decomposition fault signal.

The IE of the original signal is compared with that of each component signal, and the IE of the original signal and its components are shown in Figures 8 and 9. Notably, the IE value of IMF5, which has a dominant frequency of 2.5 MHz and exceeds the IE of the original signal, is the interference or noise portion of the original aerial-mode signal. IMF4 is chosen as the optimal component to calculate the TEO energy operator.

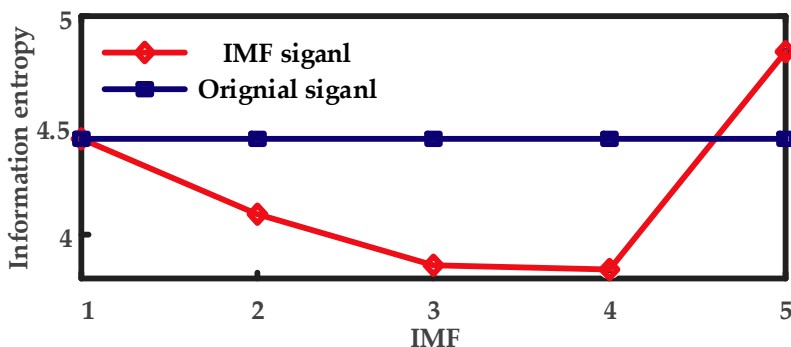

**Figure 9.** Information entropy value of the original signal and its components.

In Figure 10, the discrete TEO energy envelope of IMF4 is calculated, revealing that the aerial-mode signal arrives at the first-end detection device after 113.2 μs. The index time required for the aerial-mode signal to reach the first-terminal detection device can be determined from Equation (13): $\Delta t_1$ = 36.5 μs, $\Delta t_2$ = 94.6 μs, and $\Delta t_3$ = 133.2 μs. Since $\Delta t_2 < \Delta t$ = 113.2 μs $< \Delta t_3$, the SLG fault is located in cable section $X_2N$. According to Equation (8), the distance from injection point to the SLG fault is calculated to be 12.547 km, with an absolute error of 47 m.

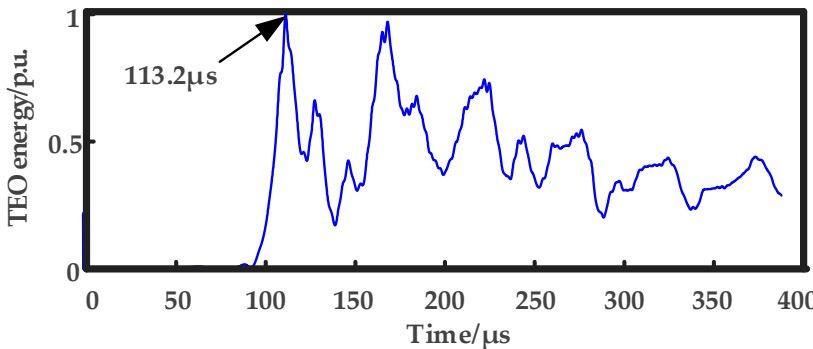

**Figure 10.** Calibration for simulated aerial-mode pulse wavefront.

### 4.3. Tests for SLG Faults of Different Location and $R_f$

To verify the reliability and location accuracy of the proposed algorithm, a practical analysis for different transition resistances $R_f$ and different sections of the hybrid feeder were conducted. The location result error was defined as $\Delta L = L_{cal} - L_{actual}$.

The effect of SLG transition resistance at 0.5 kΩ or 1 kΩ on the method proposed in this paper is verified. Various SLG faults are established for different fault locations, and the results are presented in Table 2. Whether the SLG fault occurred at the hybrid node or the end of hybrid line, the proposed scheme can locate the SLG fault with max location error less than 100 m (location error 89 m at end of $L_2$). When the fault point is located at the end of the line, the propagation distance of the aerial-mode signal is the farthest, and the serious dispersion of the aerial-mode signal leads to the largest location error.

**Table 2.** Fault location results for different fault distances.

| Fault Position | | Feeder Line | $L_{\text{actual}}$/km | $\Delta t$/μs | $L_{\text{cal}}$/km | $\Delta L$/m |
|---|---|---|---|---|---|---|
| Overhead $MX_{21}$ | | 2 | 1.500 | 11.0 | 1.512 | 12 |
| | | | 3.000 | 21.9 | 2.986 | 14 |
| Cable Line $X_{21}X_{22}$ | | 2 | 7.000 | 59.8 | 7.029 | 29 |
| | | | 8.500 | 77.2 | 8.531 | 31 |
| Overhead Line $X_{22}N$ | | 2 | 11.000 | 102.3 | 11.058 | 58 |
| | | | 14.000 | 123.4 | 13.944 | 56 |
| Hybrid node | $X_{11}$ | 1 | 6.000 | 43.8 | 6.019 | 19 |
| | $X_{21}$ | 2 | 5.000 | 36.5 | 5.020 | 20 |
| | $X_{22}$ | 2 | 10.000 | 94.6 | 10.029 | 29 |
| End of the line | | 1 | 8.500 | 72.9 | 8.549 | 49 |
| | | 2 | 15.000 | 133.2 | 15.089 | 89 |

The simulation results demonstrate that the proposed method readily achieves faulty feeder selection through the detection of aerial-mode signals at the feeder's end (Table 3). Notably, the algorithm remains unaffected by the fault point's position and can accommodate high-resistance ground faults. Moreover, the fault location outcomes are accurate and reliable, with an absolute error within 100 m. Importantly, the fault inception angles would affect the amplitude of the fault transient, thereby affecting the location accuracy. While the signal injection occurs when the hybrid distribution line is offline, the proposed method is also immune to the fault inception angle (FIA).

**Table 3.** Fault location results for different fault resistance.

| Line Type | $R_f$/kΩ | $L_{actual}$/km | $\Delta t$/μs | $L_{cal}$/km | $\Delta L$/m |
|---|---|---|---|---|---|
| Overhead Line $MX_1$ | 0.5 | 1.500 | 11.0 | 1.512 | 12 |
| | 1 | 1.500 | 11.0 | 1.512 | 12 |
| | 0.5 | 3.000 | 22.0 | 3.016 | 16 |
| | 1 | 3.000 | 22.0 | 3.016 | 16 |
| Cable Line $X_2X_1$ | 0.5 | 7.000 | 59.7 | 7.030 | 30 |
| | 1 | 7.000 | 59.7 | 7.030 | 30 |
| | 0.5 | 8.000 | 77.1 | 8.541 | 41 |
| | 1 | 8.500 | 77.1 | 8.541 | 41 |
| Overhead Line $X_2N$ | 0.5 | 11.000 | 102.4 | 11.069 | 69 |
| | 1 | 11.000 | 102.5 | 11.085 | 85 |
| | 0.5 | 14.000 | 124.5 | 14.065 | 65 |
| | 1 | 14.000 | 124.7 | 14.091 | 91 |

*4.4. Performance Comparison with Other Proposed Method*

In order to validate the proposed method for hybrid lines, a faulted-section-identification method, proposed in [12], is taken as a comparison. Using magnitudes of current traveling waves available at the line terminals, the ratios of initial aerial-mode magnitudes $\Upsilon$ are used to identify the faulted section.

Upon detecting a faulted situation, the maximum traveling waves magnitude among all modes at M-end ($i_M$) and N-end ($i_N$) are selected for the identification. The logarithm of their ratio ($i_M/i_N$) is performed and compared with the set thresholds of faulted section identification. When Equation (19) is satisfied, a fault in $k_{th}$ section is identified, where $k$ varies from 1 to *N*. The condition for identifying a fault in the $k_{th}$ section is given as follows:

$$\begin{cases} \chi_{k-1,k} = (\Upsilon_{k-1}^{max} + \Upsilon_k^{min})/2 \\ \chi_{k,k+1} = (\Upsilon_k^{max} + \Upsilon_{k+1}^{min})/2 \\ \chi_{k-1,k} < \Upsilon_f < \chi_{k,k+1} \end{cases} \tag{19}$$

where $\Upsilon_k^{max}$ and $\Upsilon_k^{min}$ represent the corresponding index maximum and minimum value for faults at $k_{th}$ junction; $\Upsilon_f$ represents the corresponding index value for the fault point; threshold *X* distinguishes the faults in the $k_{th}$ section, which can be obtained by using the parameters of the hybrid line sections for the faults that are close to the junctions.

Under the same simulation model and conditions as this paper, assuming that an SLG fault takes place somewhere in the hybrid feeders, with FIAs of 0°, 20°, 60°, or 90°, and an $R_f$ of 0.02 kΩ or 0.1 kΩ, the section determination results of the method proposed in [12] are shown in Table 4.

**Table 4.** Fault section identification results for different fault location approaches.

| Fault Section | Fault Parameters | | | $\Upsilon_f$ | Identification Result | Identified Section |
|---|---|---|---|---|---|---|
| | $R_f$/kΩ | FIA | Fault Location of Each Section/% | | | |
| Overhead Line $MX_{11}$ | 0.1 | 90° | 5 | 0.0120 | $\Upsilon < \Upsilon_{11}$ | $MX_{11}$ |
| | 0.1 | 60° | 25 | 0.0129 | $\Upsilon < \Upsilon_{11}$ | $MX_{11}$ |
| | 0.02 | 20° | 60 | 0.0136 | $\Upsilon < \Upsilon_{11}$ | $MX_{11}$ |
| | 0.02 | 0° | 95 | 0.0150 | $\Upsilon < \Upsilon_{11}$ | $MX_{11}$ |
| Cable Line $X_{11}N$ | 0.1 | 0° | 5 | 0.0153 | $\Upsilon < \Upsilon_{11}$ | $MX_{11}$ |
| | 0.1 | 20° | 25 | 0.0168 | $\Upsilon_{11} < \Upsilon$ | $X_{11}N$ |
| | 0.02 | 60° | 60 | 0.0186 | $\Upsilon_{11} < \Upsilon$ | $X_{11}N$ |
| | 0.02 | 90° | 95 | 0.0197 | $\Upsilon_{11} < \Upsilon$ | $X_{11}N$ |

| Fault Section | | R$_f$/kΩ | FIA | Fault Location of Each Section/% | $\Upsilon_f$ | Identification Result | Identified Section |
|---|---|---|---|---|---|---|---|
| | | **Fault Parameters** | | | | | |
| Overhead Line $MX_{21}$ | | 0.1 | 90° | 5 | −0.51 | $\Upsilon < \Upsilon_{21}$ | $MX_{21}$ |
| | | 0.1 | 60° | 25 | −0.46 | $\Upsilon < \Upsilon_{21}$ | $MX_{21}$ |
| | | 0.02 | 20° | 60 | −0.041 | $\Upsilon < \Upsilon_{21}$ | $MX_{21}$ |
| | | 0.02 | 0° | 95 | −0.042 | $\Upsilon < \Upsilon_{21}$ | $MX_{21}$ |
| Cable Line $X_{21}X_{22}$ | | 0.1 | 0° | 5 | −0.045 | $\Upsilon < \Upsilon_{21}$ | $MX_{21}$ |
| | | 0.1 | 20° | 25 | −0.042 | $\Upsilon < \Upsilon_{21}$ | $MX_{21}$ |
| | | 0.02 | 60° | 60 | 0.9564 | $\Upsilon_{21} < \Upsilon < \Upsilon_{22}$ | $X_{21}X_{22}$ |
| | | 0.02 | 90° | 95 | 1.0148 | $\Upsilon_{21} < \Upsilon < \Upsilon_{22}$ | $X_{21}X_{22}$ |
| Overhead Line $X_{22}N$ | | 0.1 | 90° | 5 | 1.0152 | $\Upsilon_{21} < \Upsilon < \Upsilon_{22}$ | $X_{21}X_{22}$ |
| | | 0.1 | 60° | 25 | 1.0650 | $\Upsilon > \Upsilon_{22}$ | $X_{22}N$ |
| | | 0.02 | 20° | 60 | 1.0658 | $\Upsilon > \Upsilon_{22}$ | $X_{22}N$ |
| | | 0.02 | 0° | 95 | 1.0532 | $\Upsilon_{21} < \Upsilon < \Upsilon_{22}$ | $X_{21}X_{22}$ |
| Hybrid junction | $X_{11}$ | - | - | - | $\Upsilon_{11} = 0.0154$ | - | - |
| | $X_{21}$ | - | - | - | $\Upsilon_{21} = -0.04$ | - | - |
| | $X_{22}$ | - | - | - | $\Upsilon_{22} = 1.0548$ | - | - |

The performance of the method is promising for faulted section identification when traveling wave magnitudes exceed a threshold. The method is not suitable for certain cases, such as high fault resistances at inception angles near the voltage zero crossing (*FIA* = 0°), which generate significantly lower traveling wave magnitudes. Further, when there are more sections, the traveling wave's magnitude is reduced at the measuring location due to traveling wave reflections at the junction and attenuation in the overhead and cable sections, limiting the applicability of the proposed method.

In contrast, the method proposed in this paper only needs to make use of the first arrival time of the aerial-mode component without considering the fault trigger angle; here, the $R_f$ only affects the amplitude distribution of aerial-mode component wavefront in all frequency bands at the arrival time, and the influence of the $R_f$ on the location results can be significantly reduced by selecting the arrival time of the highest-frequency component through an effective wavefront extraction algorithm.

## 5. Experiment Verification for Voltage Pulse Injection Fault Location

An experimental platform in a 10 kV distribution network (a full-scale test field) was built to verify the performance of the proposed fault-location method (as shown in Figure 11). Figure 12 presents an experimental schematic illustration. Due to experimental constraints, a section of cable with low relative wave impedance was used to simulate an overhead line.

The experimental platform comprises a pulse generator, a coupling capacitor bank [2.67 μF], a three-core armored cable (model: *R*-10-75, signal propagation velocity: $v_{11} = 150.7$ m/μs, $v_{10} = 130.5$ m/μs; model: *R*-10-50, signal propagation velocity: $v_{21} = 178.9$ m/μs, $v_{20} = 145.6$ m/μs), and three travelling wave detection devices. Three travelling wave detection devices are located at the injection point and at the end of each of the two branches. After the calculations based on the cable parameters were conducted, the parameters of the voltage pulse for the injection were determined, with a pulse amplitude of 1.5 kV and a pulse width of 0.5 μs.

A pulse was injected via the neutral point of the coupling capacitor bank when the hybrid line was cut. The SLG fault was set with the resistance of the three optional resistance values (42 Ω, 100 Ω, and 500 Ω). The actual location of the SLG fault was a distance of 1.150 km from the initial injection point. The travelling wave detection device was utilized for recording the pulse aerial-mode wavefront. The detected aerial-mode pulse wavefronts

at the end of $L_1$ and $L_2$ are shown in Figure 13. Only the aerial-mode signal can be detected at the end of the faulty feeder $L_2$, and the faulty feeder can be selected.

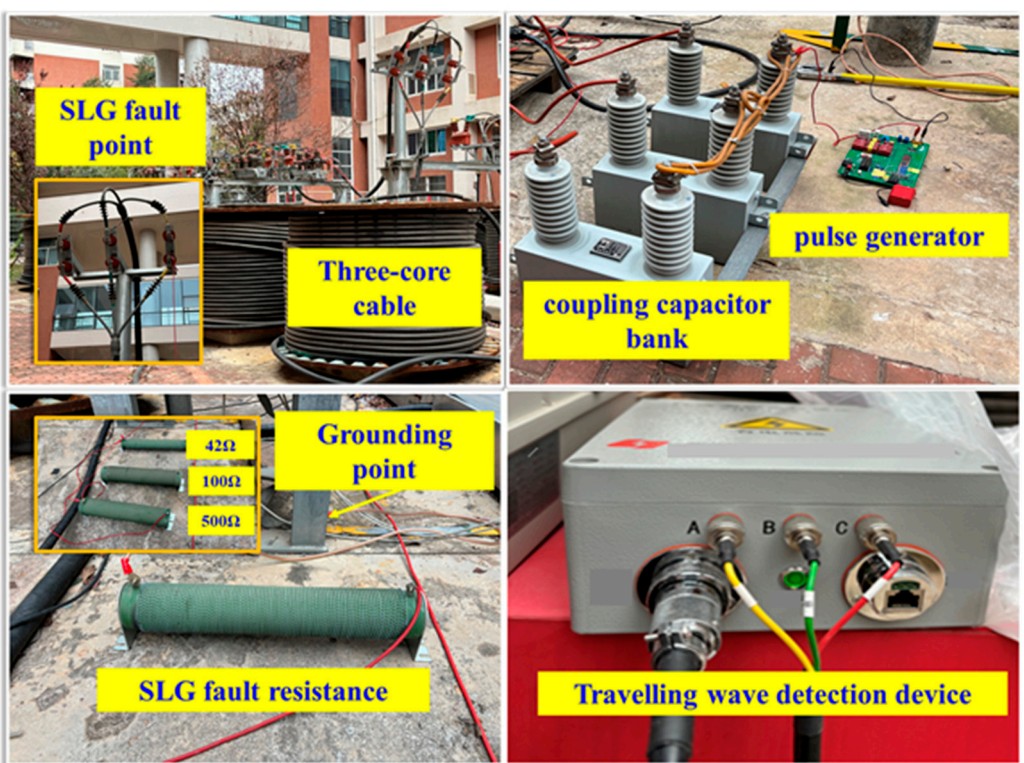

**Figure 11.** Experimental site layout.

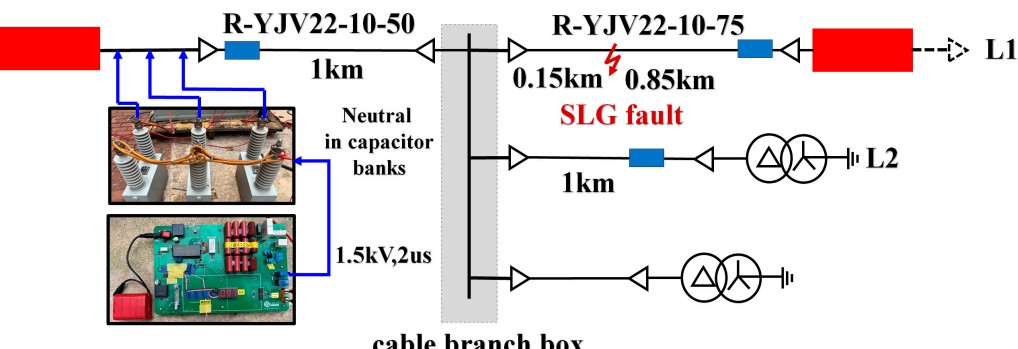

**Figure 12.** Experiment: schematic diagram.

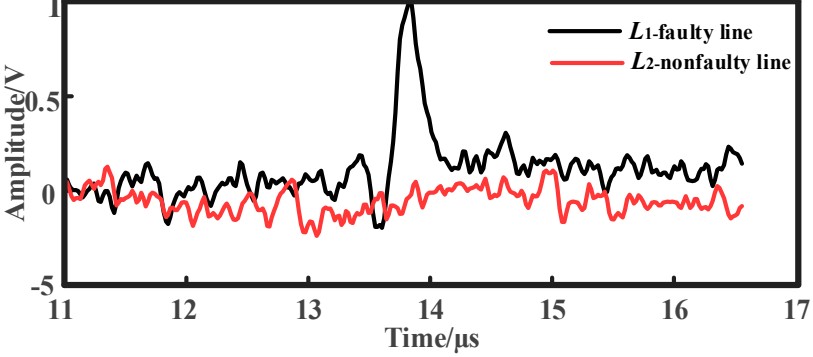

**Figure 13.** Aerial-mode pulse waveform detected at end of $L_1$ and $L_2$.

The pulse aerial-mode response under different SLG faults as shown in Figure 14. If the $R_f$ of the SLG fault increases, the amplitude of sampled aerial-mode signal decreases. The pulse aerial-mode signal collected after the fault occurs is decomposed by the VMD algorithm, and the IMFs and their corresponding spectrum are shown in Figure 15.

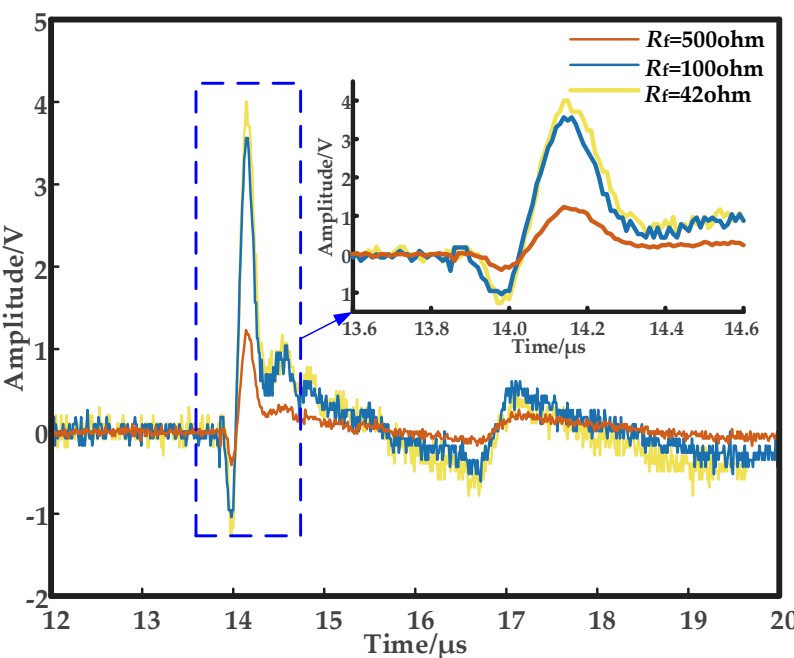

**Figure 14.** Pulse aerial-mode response under different SLG fault resistances.

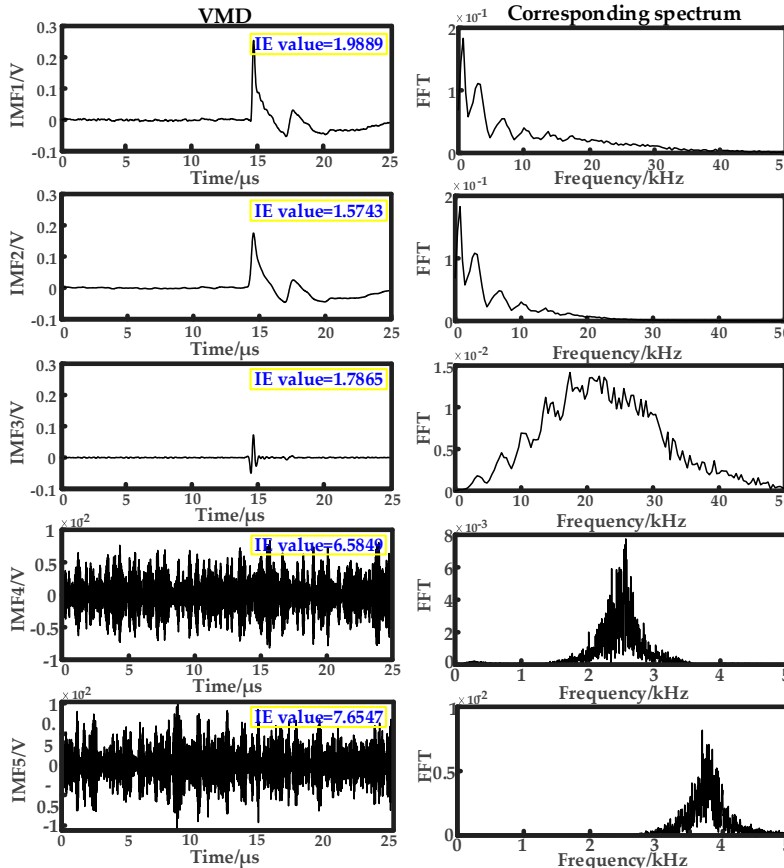

**Figure 15.** VMD decomposition fault signal and its IE value.

Then, the IE of the original signal is compared with that of each component signal. As IMF4 and IMF5 are contaminated with noise signals, the IE value of IMF4 and IMF5 exceed the IE of the original signal. The IMF3, as the second-highest-frequency component, is chosen as the optimal component for calculating the discrete TEO energy operator. According to the calibration results in Figure 16, we can determine that the aerial-mode signal arrives at the first-end detection device moment. The first pulse wavefront arrival moment is 14.5 μs. When the fault points are located at the cable branch node and at the end of $L_1$, the times needed for the aerial-mode signal to reach the first-end detection device are $\Delta t_1 = 12.5$ μs and $\Delta t_2 = 26.7$ μs, respectively. Since $\Delta t_2 < \Delta t = 14.5$ μs $< \Delta t_3$, the SLG fault is located in second cable (*R*-10-75) section. According to Equation (16), the distance from the injection point to the SLG fault is calculated to be 1.139 km, with an absolute error of 11 m.

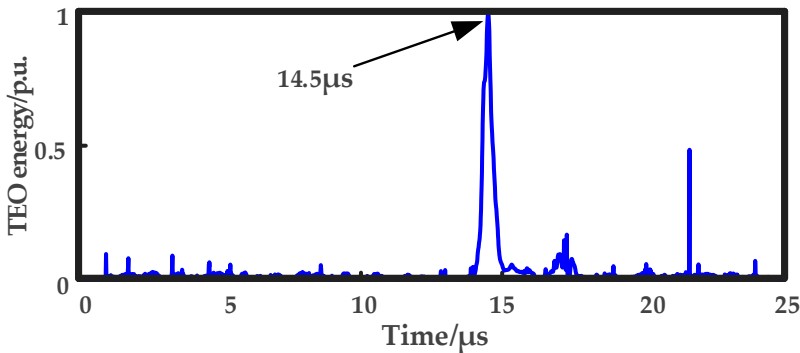

**Figure 16.** Calibration for experimental aerial-mode pulse wavefront.

## 6. Conclusions

A fault-locating scheme is proposed for hybrid distribution lines to mitigate the SLG fault location error. The complexity of the traveling wave propagation process, exacerbated by the overlapping of initial traveling wave signals, poses challenges for analysis. To address this, the scheme relies on active voltage pulse injection. By detecting the mutation of aerial-mode travelling wave after injecting pulses in all three phases simultaneously, fault information can be obtained. In terms of the branch feeder in the distribution network, the faulty line (branch) identification was completed based only on the detection of the aerial-mode signal at the end of the line. With the application of the time difference between the moment of pulse injection and the arrival moment of the aerial-mode signal at the first end of the line, it is possible to accurately locate the SLG fault in the hybrid line, adapting to any form of hybrid line configuration.

The proposed method effectively addresses the challenges associated with identifying the fault section and the reflected wave from the fault point. Unlike traditional approaches, this fault-locating scheme does not require synchronized data acquisition at both ends of the hybrid line. Additionally, there is no need to identify the fault phase in advance. The method eliminates the need for multiple pulse injections, thereby preventing damage to the lines' insulation and ensuring the safety of personnel. Theoretical and simulation results indicate that the proposed method offers higher precision than passive location methods. Experimental findings further affirm that the proposed fault-locating scheme achieves high location accuracy and convenience.

Currently, the integration of distributed generation will have a significant impact on the distribution network. The complex topology, the bidirectional and fluctuating power flow, the large number of electronic power device integrations, and the uncertainty that exists in distributed generation handling lead to significant differences in fault characteristics compared to traditional distribution networks. Whether this will affect the transmission and transformation characteristics of fault traveling waves or injected pulses is a question that it would be worthwhile to research in the future.

**Author Contributions:** Conceptualization, Z.J., F.L. and K.Y.; Data curation, Y.W.; Funding acquisition, Z.J., X.Z. and K.Y.; Methodology, Z.J. and X.Z.; Software, Z.J. and F.L.; Validation, Z.J., F.L. and L.B.; Writing—original draft preparation, Z.J.; Writing—review and editing, Z.J. All authors have read and agreed to the published version of the manuscript.

**Funding:** This work was funded by the National Natural Science Foundation of China, 52037001, Natural Science Foundation of Hunan Province, 2021JJ30729, and the Hunan Provincial Innovation Foundation for Postgraduate, CX20210790.

**Data Availability Statement:** The original contributions presented in the study are included in the article; further inquiries can be directed to the corresponding authors.

**Conflicts of Interest:** The authors declare no conflicts of interest.

## Appendix A

The cutoff frequency of a coaxial cable (also known as the critical frequency or the critical angular frequency) is the frequency below which the signal transmission within the cable is heavily attenuated and cannot effectively propagate. The cutoff frequency of a coaxial cable depends on the dimensions and materials of its inner and outer conductors. Generally, the cutoff frequency can be estimated using the following formula:

$$f_{th} = \frac{1}{2\pi\sqrt{LC}} \tag{A1}$$

where $f_{th}$ is the cutoff frequency (Hz), $L$ is the inductance per unit length of the coaxial cable (H/m), and $C$ is the capacitance per unit length of the coaxial cable (unit: F/m).

## Appendix B

The process for computing the fast Fourier transform (FFT) spectrum of pulse signals is outlined below:

1.  Definition of input signal: Let $x[n]$ represent a discrete-time domain signal, where $n$ is the discrete index of time, and $x[n]$ denotes the amplitude of the signal at time $n$.
2.  Padding of signal: If the length of the input signal is not a power of 2 (where $k$ is an integer), then it needs to be padded to the next power of 2 length, typically with zero padding. Let the length of the padded signal be $N$.
3.  FFT computation: Utilize the FFT algorithm to compute the frequency domain representation of the padded signal. The FFT algorithm transforms the signal from the time domain to the frequency domain, yielding a complex array $X[k]$, where $k$ is the discrete index of frequency, and $X[k]$ is the complex magnitude and phase of the signal at frequency $k$. The FFT formula is given by:

$$X[k] = \sum_{n=0}^{N-1} x[n] \cdot e^{-i2\pi kn/N} \tag{A2}$$

where $i$ is the imaginary unit, $N$ is the length of the signal, and $k$ is the frequency index.

4.  Calculation of amplitude spectrum: The amplitude spectrum represents the magnitude of frequencies. We can compute the amplitude spectrum, $|X[k]|$, by taking the modulus of the complex numbers.

$$X[k] = \sqrt{\text{Re}(X[k])^2 + \text{Im}(X[k])^2} \tag{A3}$$

where $\text{Re}(X[k])$ is the real part of $X[k]$ and $\text{Im}(X[k])$ is the imaginary part.

5.  Calculation of phase spectrum: The phase spectrum represents the phase of frequencies. We can compute the phase spectrum, $\angle X[k]$, by taking the arctangent of the ratio of the imaginary part to the real part.

$$\angle X[k] = \arctan\left(\frac{\text{Im}(X[k])}{\text{Re}(X[k])}\right) \tag{A4}$$

Through these steps, we can derive the frequency spectrum of the input pulse signal.

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
