# Peer review of "Active Fault-Locating Scheme for Hybrid Distribution Line Based on Mutation of Aerial-Mode Injected Pulse"

_energies, doi:10.3390/en17102248_

Round 1

Reviewer 1 Report

Comments and Suggestions for Authors

1. The use of mathematical equations (like Equation 11) needs a better context or explanation for those who might not be directly familiar with Fourier transforms and spectral analysis. Consider providing a brief explanation or referring to an appendix where these calculations are detailed step-by-step.

2. The introduction of the high-frequency component coefficient (κ) is compelling. However, it is recommended that the manuscript expounds on how this coefficient is practically calculated and how it influences the decision-making in pulse selection. A case study or a synthetic example could enhance understanding.

3. The manuscript emphasizes the safety and implementation flexibility of the proposed signal injection method, which are crucial aspects. Compare with other existing methods more explicitly, highlighting the specific advancements your approach offers in terms of operational safety and flexibility.

4. The explanation of type-A and type-B hybrid lines is clear. However, the manuscript needs more a comparative analysis of the performance of the fault location scheme on both types of lines. Are there differences in accuracy, reliability, or complexity of implementation?

5. The algorithmic approach to fault location using Δt, Δt1, Δt2, and Δt3 is mathematically intriguing. For a stronger manuscript, provide a validation of this algorithm either through simulation results or experimental setups.

6. There may be a deficiency in comparing the study’s findings with existing literature, which hampers the positioning of the research within the larger academic conversation.

7. The current number of references, 23, may be perceived as low. Consider expanding the reference base to demonstrate a comprehensive understanding of the topic and to validate your research through existing literature.

8. Ensure that the references cited are recent and encompass a wide range of perspectives. An up-to-date and extensive reference list will strengthen the credibility of your research and demonstrate a thorough review of existing literature.

9. Povide a clear and detailed explanation of the verification process used in the simulation to ensure the accuracy and reliability of the results. Consider including validation against known data or a benchmark to enhance the credibility of the simulation outcomes.

10. The paper requires strengthening the theoretical foundation of hybrid distribution lines and demonstrates an understanding of advanced communication techniques in the context of fault location schemes. Additionally, it showcases a comprehensive approach to addressing challenges in fault location methods, aligning with the research objectives and enhancing the scholarly credibility of the paper.

https://doi.org/10.1007/s11082-020-02497-0

Comments on the Quality of English Language

Moderate revisions. 

Reviewer 2 Report

Comments and Suggestions for Authors

Main remarks
- Line 202 – A more extensive comment is required: why the signal bandwidth can exceed 100kHz?
- Line 247 Figure 2 Nestling a figure within another figure can make it difficult to read. The figure should be redrawn.
- In the Conclusion section (Line 469), there is a lack of outlining directions for future research. This needs to be added. Do the authors plan to conduct comprehensive studies to assess impact of environmental factors such as weather conditions, electromagnetic interference, and line impedance variations on the propagation characteristics of aerial-mode waves and their implications for fault location accuracy? If so, a description of future research could be beneficial for readers.

Other minor remarks and recommendations
- Could the Authors comment on the promotion of standardization and guidelines for aerial-mode traveling wave technology to ensure interoperability and widespread adoption in relation to the proposed solution in the manuscript?

Round 2

Reviewer 1 Report

Comments and Suggestions for Authors

The paper has been revised appropriately, with significant improvements made. It is now considered acceptable. This work represents an excellent effort.

Final recommendation: Accept 

Comments on the Quality of English Language

Minor editing.